# Uptake and User Characteristics for Pharmacy-Based Contraception and Chlamydia Treatment: A Quantitative Retrospective Study from the UK

**DOI:** 10.3390/pharmacy9010061

**Published:** 2021-03-17

**Authors:** Julia Gauly, Helen Atherton, Jonathan D. C. Ross

**Affiliations:** 1Warwick Medical School, University of Warwick, Coventry CV4 7HL, UK; H.Atherton@warwick.ac.uk; 2University Hospitals Birmingham NHS Foundation Trust, Birmingham B15 2TH, UK; Jonathan.Ross@uhb.nhs.uk

**Keywords:** contraception, chlamydia treatment, sexual health

## Abstract

The health provider Umbrella delivers several SRHS through more than 120 pharmacies in Birmingham (England). Umbrella pharmacy data collected between August 2015 and August 2018 were used to descriptively analyse the uptake and user characteristics for emergency contraception, short-acting oral contraception, condoms and chlamydia treatment. In total, 54,309 pharmacy visits were analysed. A total of 30,473 females presented for emergency contraception. Most were supplied with an emergency contraceptive pill (98.6%, 30,052 out of 30,473), which was levonorgestrel in 57.4% of cases (17,255 out of 30,052). Of those females who attended for short-acting oral contraception, 54.3% (1764 out of 3247) were provided with the progesterone-only pill. Of those who were given chlamydia treatment, the majority received doxycycline (76.8%, 454 out of 591). A total of 74% (14,888 out of 19,998) of those who requested condoms were not provided with specific instructions on their use. Pharmacies have the potential to make a substantial contribution to the delivery of an integrated sexual health service including rapid access to emergency contraception, convenient delivery of short-acting hormonal contraception and treatment of chlamydia. Appropriate education, support and audit is required to ensure the delivery of high-quality care.

## 1. Introduction

Pharmacies are the most frequently visited health care destination and play a vital part in the delivery of health care worldwide [1]. Policy makers are increasingly recognising the importance of extending pharmacy staff roles to meet growing health care demands. Several countries including Australia, Canada, New Zealand and the United States are funding public health services in pharmacies [2,3,4]. Recent policies aim to extend pharmacists’ practice beyond dispensing into more clinical and patient-centred roles [5]. The expansion of pharmacists’ role has been promoted by the WHO [6].

The United Kingdom (UK) has been at the forefront of expanding pharmacists’ roles, and many pharmacies in the UK provide public health services such as smoking cessation, lifestyle advice, substance use management and sexual and reproductive health services (SRHS) [2]. SRHS has been identified as one area in which pharmacies could be more involved.

Since 2013, these services in England have been mainly commissioned by local authorities. The changes in the delivery of SRHS have been challenging but also provided a unique opportunity for SRHS to be re-designed. Integrated services incorporating contraception and sexually transmitted infections (STIs) have become more common, meaning that people only need to visit one site to obtain a range of care. With local authorities often procuring SRHS using tender processes, many different providers, including pharmacies, have become involved in providing services. However, the provision of pharmacy-based SRHS varies widely across the UK and little evaluation of pharmacy-based SRHS is available to date, making it difficult to understand pharmacies’ contribution to the delivery of services. The recent report of the ‘All-Party Parliamentary Group (APPG) on Sexual Health and Reproductive Health in the UK’ highlighted the importance of collecting and reporting data on the uptake and user characteristics of SRHS for assessing access and efficient planning and commissioning [7]. 

In Birmingham, the health provider Umbrella delivers a wide range of SRHS in more than 120 pharmacies [8]. In the current study, we analysed the uptake and user characteristics for pharmacy-based emergency contraception, short-acting oral contraception, condoms and chlamydia treatment over a three-year time period (August 2015 to August 2018). 

## 2. Materials and Methods

The information governance team of the University Hospitals Birmingham (UHB) NHS Foundation Trust set criteria for processing the information to ensure patient anonymity was maintained. Ethical approval was received from the South Central—Oxford B Research Ethics Committee and from the Health Research Authority (REC Reference: 18/SC/0511). Additionally, local approval from the University Hospital Birmingham NHS Foundation Trust was obtained.

### 2.1. Umbrella Pharmacy Services

Umbrella’s pharmacy services are accessible via self-referral to those aged over 13 years old and are free of charge to the user. Umbrella uses Patient Group Directions (PGDs), a mechanism that allows the prescription-free delivery of SRHS which would usually require users to see a prescribing clinician to provide short-acting oral contraception and chlamydia treatment. All pharmacy staff delivering SRHS are trained by Umbrella on communication skills with young people, confidentiality, condom provision and signposting. There are two different levels of Umbrella pharmacies services, Tier 1 and, for more complex management, Tier 2. Staff in both Tier 1 and Tier 2 pharmacies receive training to deliver emergency contraception (EC) and condoms. Short-acting oral contraception and chlamydia treatment are only provided in Tier 2 pharmacies. In order to receive chlamydia treatment from an Umbrella pharmacy, no proof of a positive test result is required, and it is not recorded whether and from which source a chlamydia test was obtained by people presenting for chlamydia treatment. Hormonal contraception (EC and short-acting oral contraception) is only available for those whose biological sex is female, whereas chlamydia treatment and condoms are provided to people of both sexes. Pharmacy users who attend for an Umbrella service in the pharmacy are seen in a private pharmacy consultation room by a trained member of pharmacy staff. An electronic patient record containing demographic, and management information is completed by pharmacy staff during the consultation which is conducted by a trained pharmacist for most services (including emergency contraception, oral contraception, and chlamydia treatment). However, pharmacy health care assistants who are trained by Umbrella can deliver some SRHS (e.g., condoms).

Females presenting for emergency contraception are offered an emergency contraceptive pill (either levonorgestrel or ulipristal) and/or insertion of a copper coil. For females who, after counselling, decide to have a copper coil inserted, pharmacy staff can either provide a referral to or book an appointment with a sexual health clinic in Birmingham. If pharmacy staff conclude, based on information provided during the consultation, that females are either not at risk of getting pregnant or that it is too late to provide emergency contraception, females may be provided with advice only. Hence, at the end of a consultation on emergency contraception, females can be provided with: An emergency contraceptive pill (either levonorgestrel or ulipristal) only,An emergency contraceptive pill and arrangements for insertion of a copper coil at a sexual health clinic, An arranged insertion of a copper coil at a sexual health clinic, andAdvice only.

Short-acting oral contraception can be initiated to first-time pill users or to established pill users. Females receiving short-acting oral contraception are either provided with the progesterone-only pill or the combined pill at the end of a consultation. Males and females presenting for chlamydia treatment are provided with either doxycycline or azithromycin. Pharmacy users requesting condoms are provided with sexual health advice and asked whether or not they would like to also receive written instructions on condom use.

### 2.2. Data Source

Umbrella collects data for their pharmacy services and is authorised to use anonymised data for any legitimate purpose. Data on Umbrella’s pharmacy services is collected by pharmacy staff using PharmOutcomes^®^ (Pinnacle Health Partnership LLP, East Coves, England), a secure web-based system which is used by 85% of pharmacies in England as an electronic patient record [9]. PharmOutcomes^®^ data on Umbrella’s services were analysed for the time period August 2015 to August 2018. When asked for their gender, users have the possibility to self-identify as female, male, or transgender.

### 2.3. Data Analysis 

A descriptive analysis was conducted using the following variables: 

User demographics: Gender,Age group, andEthnicity.

Uptake: EC,Short-acting oral contraception,Chlamydia treatment, andCondoms.

The data analysis was conducted using IBM^®^ SPSS Statistics software (SPSS^®^) version 24 [10]. 

## 3. Results

The uptake and user characteristics of pharmacy-based SRHS are presented in Table 1. In total, 54,309 pharmacy visits made for an Umbrella service were analysed. 

For all services (emergency contraception, short-acting oral contraception, condoms, chlamydia treatment), service users were most likely to be female, White/White British and between 20 and 24 years old. 

The large majority of females presenting for EC were supplied with an emergency contraceptive pill (98.6%), of whom 57.4% received levonorgestrel and 42.6% received ulipristal. Few females were referred for insertion of an emergency copper coil (<1%) or documented to have been only offered advice on emergency contraception (<1%). 

Slightly more females requesting short-acting oral contraception were provided with the progesterone-only pill (54.3%) than with the combined pill (44.4%). Most people attending for chlamydia treatment received doxycycline (76.8%) compared to azithromycin (20%). For those requesting condoms, few were documented as having received specific instructions on their use (25.6%).

## 4. Discussion

The All-Party Parliamentary Group on Sexual Health and Reproductive Health has stressed the importance of reporting data on SRHS and of expanding the use of PGDs to improve access to prescription-only SRHS in pharmacies [7]. Our analysis of data from a large pharmacy service which provides SRHS through PGDs helps to inform future service delivery.

In line with a recent study from Australia [11], our analysis showed that females’ contraceptive responsibility was not necessarily tied to female-specific methods, with females accounting for the majority of access for all contraceptive methods, including condoms. This may indicate that females take responsibility for any type of protection. However, while males can attend for condoms, access to hormonal contraceptive methods is restricted to females only in Umbrella pharmacies. Further, following Umbrella’s integrated sexual health model, it is likely that females presenting for hormonal contraception were additionally offered condoms by pharmacy staff as protection from STIs. Nevertheless, the data show that the uptake of condoms by males is relatively low. This is despite a recent survey from the UK which showed that pharmacies are males’ most popular choice for obtaining contraception [12]. It is possible that some males bought condoms in the pharmacy rather than requesting them for free through the Umbrella service. This may have been because they were unaware of the Umbrella service, or that they did not want to disclose personal information or ask for condoms at the counter [13]. However, it is also possible that males did not obtain condoms because they did not want to use them or because they did not think they were at risk of contracting an STI [14]. To better understand the uptake of condoms in pharmacies, the number of condoms sold by pharmacies and provided free via the Umbrella service should be compared in a future study.

In line with a previously published study, the majority of EC consultations ended in the supply of an EC pill containing levonorgestrel [15]. EC pills containing levonorgestrel are licensed for use up to 72 h following unprotected sexual intercourse, whereas EC pills containing ulipristal can also be provided between 72 and 120 h after UPSI [15]. Our finding suggests that most females present in pharmacies within 72 h after unprotected sexual intercourse.

According to Umbrella, each woman presenting for EC in an Umbrella pharmacy should be offered an EC pill and/or an emergency copper coil insertion. Our analysis showed that less than 1% of females attending for EC were referred for insertion of a copper coil. While the reasons for this were not captured, this may be due to concerns about the prolonged efficacy and side-effects of intrauterine contraception [16] or because of lack of appointments for coil insertion [13]. A recent clinic-based study showed that females who presented within 72 h were often not offered the copper coil, but that the proportion of females offered both an IUD and an EC pill increased with longer times after unprotected sexual intercourse [17]. Although pharmacy staff had been trained to inform patients about the copper coil, it is also possible that this may not happen in practice. 

We found that instructions on use of condoms were rarely provided and that females were less likely to be given instructions on use of condoms. Although condom instructions may contribute to safer sex, a recent study from the US showed that discussing the use of condoms can make people feel embarrassed [18] and it is possible that people declined the condom advice because of this and this led to the low number of condom instructions being provided. However, it is also possible that pharmacy staff did not accurately record whether instructions were provided or not. Females are more likely than males to utilise health resources [19,20] and possibly as a consequence, a recent study showed that females had superior condom application and removal skills [14]. It is possible that more females than males declined instructions as they had previously obtained information on the correct use of condoms. However, it could also be that females did not feel it was their responsibility to know how to use a condom. More research on females’ and males’ condom use skills and preferred sources of knowledge on condom usage should be conducted.

The majority of patients with chlamydia were treated with doxycycline which is consistent with current guidance [21]. 

In contrast to a study on a specialist contraceptive service from the UK, most females in our study were provided with the progesterone-only pill rather than the combined pill [22]. It is possible that the progesterone-only pill was supplied more commonly because it has less contra-indications and because it is easier to train pharmacy staff to supply it safely. In contrast to combined hormonal contraception, progesterone-only pills do not increase the risk of venous thromboembolic events (VTE). As the risk of VTE increases with age [23], this may explain why females over the age of forty were more frequently provided the progesterone-only pill.

This study has a number of potential limitations: The absence of consistent patient identification numbers meant that an analysis was only possible for each service provided and not at an individual patient level, and some patients may have attended on multiple occasions over the study period. It was also unclear from the electronic patient record how closely pharmacists adhered to the PGDs when offering advice and onward referral, for example, whether not providing condom advice was due to the patient declining or it not being discussed.

The uptake of services and patient outcomes following a pharmacy consultation can inform the design of integrated SRHS and inform future audit and training needs of pharmacists and ancillary pharmacy staff. 

## Figures and Tables

**Table 1 pharmacy-09-00061-t001:** Uptake and user characteristics of pharmacy-based SRHS.

Emergency Contraception		Emergency Contraceptive Pill Only	Emergency Contraceptive Pill and Copper Coil Appointment/Referral	Advice Only	Copper Coil Appointment/Referral Only	Unknown ^1^
Total (% by column)		29,961 (100%)	91 (100%)	34 (100%)	9 (100%)	378 (100%)
Gender (% by column)	Female	29,952 (100%)	91 (100%)	34 (100%)	9 (100%)	378 (100%)
	Other ^1^	9 (0%)	-	-	-	-
Ethnicity (% by column)	White/White British	12,187 (40.7%)	39 (42.9%)	17 (50%)	4 (44.4%)	136 (36.0%)
	Asian/Asian British	7358 (24.6%)	23 (25.3%)	6 (17.6%)	1 (11.1%	90 (23.8%)
	Black/Black British	4863 (16.2%)	12 (13.2%)	4 (11.7%)	1 (11.1%)	85 (22.5%)
	Mixed/Mixed British	2093 (7.0%)	8 (8.8%)	5 (14.7%)	3 (33.3%)-	24 (22.5%)
	Other ethnic group	555 (1.9%)	6 (6.6%)	-	-	10 (2.6%)
	Unknown	2906 (9.7%)	3 (3.3%)	2 (5.9%)	-	33 (8.7%)
Age (% by column)	13–15	266 (0.9%)	1 (1.1%)	-	-	-
	16–19	6145 (10.5%)	20 (22.0%)	7 (20.6%)	1 (11.1%)	50 (13.2%)
	20–24	10,583 (35.3%)	35 (38.5%)	17 (50%)	5 (55.6%)	100 (26.5%)
	25–29	6085 (20.3%)	19 (20.9%)	4 (11.8%)	1 (11.1%)	100 (26.5%)
	30–39	5586 (18.6%)	12 (13.2%)	5 (14.7%)	2 (22.2%)	110 (29.1%)
**Short-Acting Oral Contraception**		**Progesterone-Only Pill**	**Combined Pill**	**Unknown ^1^**		
Total (% by column)		1764 (100%)	1442 (100%)	41 (100%)		
Gender (% by column)	Female	1762 (100%)	1442 (100%)	41 (100%)		
	Other	2 (0%)	-	-		
Ethnicity (% by column)	White/White British	1078 (61.1%)	813 (56.4%)	27 (65.9%)		
	Asian/Asian British	203 (11.5%)	242 (16.8%)	5 (12.2%)		
	Black/Black British	251 (14.2%)	171 (11.9%)	5 (12.2%)		
	Mixed/Mixed British	87 (4.9%)	85 (5.9%)	1 (2.4%)		
	Other ethnic group	19 (1.1%)	29 (2.0%)	2 (4.9%)		
	Unknown	-	102 (7.1%)	1 (2.4%)		
Age (% by column)	13–15	2 (0.1%)	17 (1.2%)	-		
	16–19	276 (15.6%)	256 (17.8%)	10 (24.4%)		
	20–24	536 (30.4%)	608 (42.2%)	12 (29.3%)		
	25–29	296 (16.8%)	339 (23.5%)	10 (24.4%)		
	30–39	398 (14.5%)	203 (14.1%)	7 (17.1%)		
	40+	256 (14.5%)	19 (1.3%)	2 (4.9%)		
**Chlamydia Treatment**		**Azithromycin**	**Doxycycline**	**Unknown ^1^**		
Total (% by column)		118 (100%)	454 (100%)	19 (100%)		
Gender (% by column)	Female	81 (68.6%)	288 (63.4%)	13 (68.4%)		
	Male	37 (31.4%)	163 (35.9%)	6 (31.6%)		
	Other ^2^	-	3 (0%)	-		
Ethnicity (% by column)	White/White British	60 (50.8%)	210 (46.3%)	9 (47.4%)		
	Asian/Asian British	9 (7.6%)	21 (4.6%)	-		
	Black/Black British	21 (17.8%)	78 (17.2%)	6 (31.6%)		
	Mixed/Mixed British	8 (6.8%)	47 (10.4%)	2 (10.5)		
	Other ethnic group	3 (2.5%)	5 (1.1%)	1 (5.3%)		
	Unknown	-	93 (20.5%)	1 (5.3%)		
Age (% by column)	16–19	29 (24.6%)	86 (18.9%)	5 (26.3%)		
	20–24	51 (43.2%)	203 (44.7%)	7 (36.8%)		
	25–29	17 (14.4%)	104 (22.9)	6 (31.6%)		
	30–39	15 (12.7%)	47 (10.4%)	-		
**Condoms**		**Instructions Provided**	**No Instructions Provided**	**Unknown ^1^**		
Total (% by column)		1331(100%)	14888 (100%)	3779 (100%)		
Gender (% by column)	Female	800 (60.1%)	10082 (67.7%)	2404 (63.6%)		
	Male	523 (39.3%)	4733 (31.8%)	1354 (35.8%)		
	Other ^2^	8 (0.6%)	73 (0.4%)	21 (0.6%)		
Ethnicity (% by column)	White/White British	494 (37.1%)	6719 (45.1%)	1790 (47.4%)		
	Asian/Asian British	361 (27.1%)	3384 (22.7%)	867 (22.9%)		
	Black/Black British	182 (13.7%)	1994 (13.4%)	539 (14.3%)		
	Mixed/Mixed British	80 (6.0%)	854 (5.7%)	175 (4.6%)		
	Other ethnic group	52 (3.9%)	300 (2.0%)	77 (2.0%)		
Age (% by column)	13–15	32 (2.4%)	206 (1.4%)	58 (1.5%)		
	16–19	223 (16.8%)	2634 (17.7%)	850 (22.5%)		
	20–24	310 (23.3%)	4483 (30.1%)	1232 (32.6%)		
	25–29	213 (16%)	2196 (14.8%)	552 (14.6%)		

^1^ Missing data—outcome of SRHS was not recorded by pharmacy staff in Umbrella pharmacy for unknown reason. ^2^ Unknown includes people who self-identified as transgender and entries where the gender was not recorded by pharmacy staff.

## Data Availability

The data contained in this article was obtained by Umbrella, who owns that data. The data is not publicly available.

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
