# Peer review of "Uptake and User Characteristics for Pharmacy-Based Contraception and Chlamydia Treatment: A Quantitative Retrospective Study from the UK"

_pharmacy, 2021, doi:10.3390/pharmacy9010061_

Round 1

Reviewer 1 Report

Thank you for the opportunity to review the manuscript “Uptake and user characteristics for pharmacy-based contraception and chlamydia treatment: A quantitative retrospective study from the UK”.

The manuscript is very well written, and structured, reporting on the total uptake for the Umbrella pharmacies (a concept that is nicely introduced) in Birmingham during 3 years.

Having said that, some features that are missing in the manuscript:

Lack of international literature in the Introduction – what do we know from earlier relevant practices of this kind in pharmacies? I.e. put this manuscript into an international context.

At the end of the manuscript, something seems to be missing (part of Discussion and Conclusion?) - anyway needs to be included. Hence it might be that my next comment is actually already answered – but as I can’t see it: Why are socio-demographics (gender, ethnicity and age) included when not taken into account in e.g. the discussion? For example, more women than men get condoms but a larger proportion of women than men do not get instructions. (Are women rather than men getting condoms a sign of women (still) taking the main responsibility for protection? Are men uncomfortable with visiting pharmacies? What does the literature say?) This is just one example of interesting data that you are not commenting on.

In this sense your aim - to analyze the data – not fulfilled, rather you present the data. So, what do we know about gender, ethnicity and age when it comes to sexuality et c. – and how can that be related to your data.

Author Response

Many thanks for taking the time to review our paper and for your valuable feedback – we very much appreciate it. We have addressed all of your comments  as outlined in the attachment. 

Reviewer 2 Report

This manuscript titled, "Uptake and user characteristics for pharmacy-based contraception and chlamydia treatment: A quantitative retrospective study from the UK" analyzes the pharmacy services that provides Sexual and Reproductive Health Services (SRHS) in the UK.  

Overall, the authors have done a good job of analyzing the beneficiary services of SRHS at the pharmacies. Authors have also clearly mentioned the limitations of their study such as lack of patient identifiers to examine services on an individual level and some of the pharmacies advice/referral process. 

Minor comments:

In Page# 7 - The last sentence in the discussion section is incomplete. Please correct it.

Author Response

Many thanks for taking the time to review our paper and for your positive feedback – we very much appreciate it.

Many thanks for noticing that the final sentence was incomplete – we have completed the last sentence in the discussion section as shown in the attachment. 

Round 2

Reviewer 1 Report

-

Author Response

Many thanks for taking the time to review our corrections. We have proofread the article again and corrected any minor mistakes, e.g.: 

Abstract, line 15: In total, 54 309 pharmacy visits were analysed. 30 473 females presented for emergency contraception.

4. Discussion, Line 201: Our finding suggests that most females present in pharmacies within 72-hours after unprotected sexual intercourse.